# SuperSNN: Training Spiking Neural Networks with Knowledge from Artificial Neural Networks

## Abstract

Spiking Neural Network (SNN) is a kind of brain-inspired and event-driven network, which is becoming a promising energy-efficient alternative to Artificial Neural Networks (ANNs). However, the performance of SNNs by direct training is far from satisfactory. Inspired by the idea of Teacher–Student Learning, in this paper, we study a novel learning method named *SuperSNN*, which utilizes the ANN model to guide the SNN model learning. *SuperSNN* leverages knowledge distillation to learn comprehensive supervisory information from pre-trained ANN models, rather than solely from labeled data. Unlike previous work that naively matches SNN and ANN's features without deeply considering the precision mismatch, we propose an indirect relation-based approach, which defines a pairwise-relational loss function and unifies the value scale of ANN and SNN representation vectors, to alleviate the unexpected precision loss. This allows the knowledge of teacher ANNs can be effectively utilized to train student SNNs. The experimental results on three image datasets demonstrate that no matter whether homogeneous or heterogeneous teacher ANNs are used, our proposed *SuperSNN* can significantly improve the learning of student SNNs with only two time steps.

## 1 Introduction

Spiking Neural Network (SNN) is a kind of biologically plausible neural network based on dynamic characteristics of biological neurons (Mcculloch & Pitts, 1943; Izhikevich & E., 2003). Previous research has demonstrated the potential of SNNs in achieving energy savings while enabling fast inference (Stckl & Maass, 2020). However, the performance of SNNs is still far from satisfactory. Although surrogate gradient methods (Yujie et al., 2018; Shrestha & Orchard, 2018) have been proposed to realize the direct training of SNNs, they often result in SNNs with lower accuracy and slower convergence rates compared to ANNs.

In ANNs, Teacher-Student (T-S) learning (Manohar et al., 2018) is a transfer learning approach, providing comprehensive supervisory information from the teacher model to guide the student model for better performance of learning. Enlightened by the idea of Teacher-Student learning, a question arises: Can we enhance the performance of SNNs by learning knowledge of ANNs? Unlike ANN-ANN learning in which knowledge is transferred using the same knowledge representation, ANN-SNN learning transfers knowledge between two kinds of knowledge representation, which brings two main challenges. The first is that compared with ANN, SNN has an additional dimension of knowledge - temporal to convey, leading to the dimension mismatch between the representation vectors of ANNs and SNNs. The second is that the neuron state of ANNs is represented in binary format but that of SNNs is represented in float format, leading to the precision mismatch between ANNs and SNNs.

Considering the above challenges, in this paper, we propose a novel T-S learning approach for SNN, named *SuperSNN*, which can effectively reduce the representation mismatch between ANN and SNN. As a relation-based approach of knowledge distillation, *SuperSNN* directs SNN learning with a pairwise-relational loss, helping the semantic relationship of the knowledge learned by ANNs be well preserved and transferred to SNN. To overcome the difficulty caused by the dimension mismatch, inspired by the work (Xu et al., 2023b), we exploit the idea of average pooling over

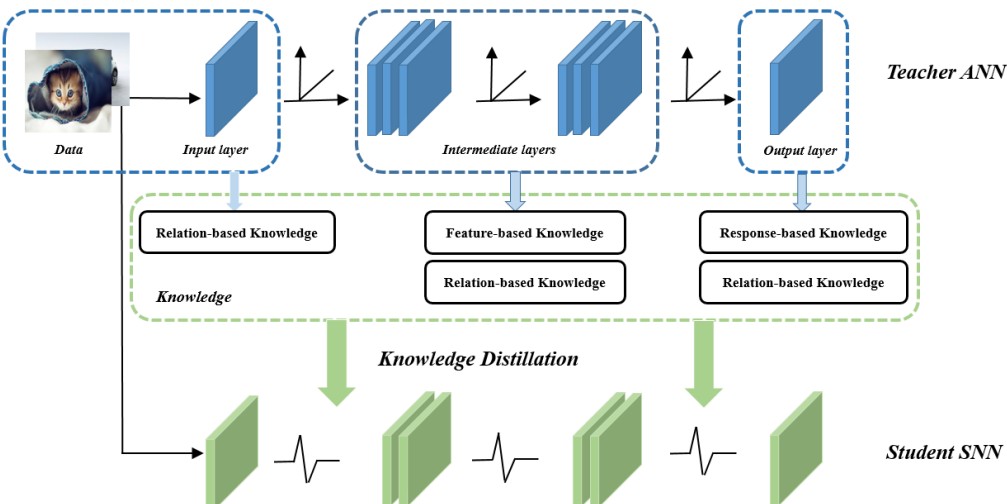

Figure 1: The illustration describes how a student SNN learns response-based knowledge, feature-based knowledge, and relation-based knowledge from a teacher ANN. Typically, response-based and feature-based knowledge are obtained solely from the teacher ANN's output layer and intermediate layers. But, relation-based knowledge can be derived from all the layers with multi-level conceptual features. As a result, relation-based knowledge distillation is more flexible and easy to use.

SNN's temporal dimension to eliminate the extra feature, making SNN and ANN have the same feature dimensions. Additionally, unlike other work (Xu et al., 2023b) that match the representation vectors of SNN and ANN with little concern about the precision mismatch, we present an indirect matching approach, which unifies the value scale of SNN and ANN representation vectors and define a pairwise-relational loss function, alleviating the unexpected and unnecessary loss of precision. Furthermore, as shown in Figure 1, compared to response-based approaches (Kushawaha et al., 2021; Xu et al., 2023a) and feature-based approaches(Xu et al., 2023b; Yang et al., 2022), *SuperSNN* is more flexible and easy-to-use, which can be applied to every layer of the networks. To the best of our knowledge, it is the first time to explored the relation-based approach in ANN-SNN learning.

At last, to show the effectiveness of *SuperSNN*, we choose ResNet18 and Pyramidnet50 as SNN models and compare *SuperSNN* with current leading SNN approaches over three benchmarks (CIFAR10, CIFAR100, Tiny ImageNet). The experiments show that no matter whether homogeneous or heterogeneous teacher ANNs are used, the proposed *SuperSNN* can outperform other SNN training methods using only two time steps, which proves the reliability and validity of *SuperSNN*.

## 2 BACKGROUND

**Spiking Neuron Networks.** Unlike traditional ANNs, SNNs use binary spike trains to transmit information. Here we use the iterative Leaky Integrate-and-Fire (LIF) neuron (Yujie et al., 2018) as the basic neuron model of student SNNs. When the membrane potential exceeds a specific threshold, the neuron fires a spike and the membrane potential will be reset to zero. The whole iterative LIF model in both spatial and temporal domains can be determined by

$$u^{t,l+1} = \tau u^{t-1,l+1}(1 - o^{t-1,l+1}) + x^{t,l} \tag{1}$$

$$o^{t,l+1} = \begin{cases} 1 & \text{if } u^{t,l+1} > \theta^{l+1} \\ 0 & \text{otherwise} \end{cases} \tag{2}$$

where $u^{t,l}$ is the membrane potential of the neuron in $l$-th layer at time $t$, $o^{t,l}$ is the binary spike. $\tau$ represents the membrane time constant, a constant to describe how fast the membrane decays, $x^{t,l}$ denotes the external input current, which comes from the weighted sum of the spikes fired by the

neurons in $l$-th layer. $\theta^l$ is the threshold in $l$-th layer. In conclusion, the iterative LIF model enables forward and backward propagation to be implemented on both spatial and temporal dimensions.

**Notations.** ANNs are good at learning multiple levels of feature representation with increasing abstraction (Bengio et al., 2013). Therefore, not only the output of the last layer (Hinton et al., 2015) but also the outputs of intermediate layers (Romero et al., 2015) can be extracted as the knowledge sources to supervise the training of student SNNs. Such layer outputs are so called *feature maps*. (Gou et al., 2021),

Denote a teacher ANN as $\mathcal{T}$ and a student SNN as $\mathcal{S}$, For an input mini-batch, let the *feature map* of $\mathcal{T}$ at the layer $l$ be $A_{\mathcal{T}}^l \in \mathbb{R}^{B \times C \times H \times W}$, where $B$, $C$, $H$, $W$ are the batch size, channel number, height, width respectively. As mentioned before, the intermediate output of SNN contains an additional temporal dimension, thus the *feature map* of $\mathcal{S}$ at the layer $l'$ is defined as $A_{\mathcal{S}}^{l'} \in \mathbb{R}^{B \times T \times C' \times H' \times W'}$, where $C'$, $H'$, $W'$, $T$ are the number of channels, height, width, time step respectively.

## 3 OUR APPROACH

In this section, we will give our novel learning method named *SuperSNN*, which utilizes the ANN model to guide SNN model learning. In this section, we first introduce a pairwise-relational knowledge. Based on this knowledge definition, a pairwise-relational loss function is well-designed. At last, the overall training process is fully described.

### 3.1 PAIRWISE-RELATIONAL KNOWLEDGE

Considering the mismatch problems mentioned before, we believe relation-based knowledge is more suitable for ANN-SNN transfer learning. Because relational distances are mainly presented to model the relative position of two features rather than their exact distance value. Therefore, relation-based knowledge is naturally friendly to ANN-SNN transfer learning, where the exact distance between SNN's binary features and ANN's float features is hard to compute precisely.

In our approach, to model relational knowledge, we use *pairwise similarities* (Tung & Mori, 2019) to represent the relational distance between two knowledge features. As shown in Figure 2, SNN has one more dimension of time than ANN, for generality, we exploit a classical method, average pooling (Xu et al., 2023b), to get rid of the time dimension. After the process of average pooling over the time dimension, SNN's feature map will transfer from $A_{\mathcal{S}}^{l'} \in \mathbb{R}^{B \times T \times C' \times H' \times W'}$

Figure 2: Illustration of calculating new representations of SNNs based on *feature maps* from intermediate layers. Due to the binary spike in SNNs, an additional time dimension $T$ is introduced to transfer time information. Therefore, to align the feature map size of SNNs and that of ANNs, the values along the time dimension are averaged.

to $A_{\mathcal{S}'}^{l'} \in \mathbb{R}^{B \times C' \times H' \times W'}$, which has the same dimension with ANN's feature map $A_{\mathcal{T}}^l \in \mathbb{R}^{B \times C \times H \times W}$.

As shown in Figure 3, for ease of calculation, we simply reshape the *feature map* $A_{\mathcal{T}}^l$ and $A_{\mathcal{S}'}^{l'}$ into $\mathbb{R}^{B \times CHW}$ and $\mathbb{R}^{B \times C'H'W'}$, and formally define the *pairwise similarities* on the model $\mathcal{T}$ and $\mathcal{S}$ as:

$$\widetilde{Q}_{\mathcal{T}}^l = A_{\mathcal{T}}^l \cdot {A_{\mathcal{T}}^l}^{\top}; \quad Q_{\mathcal{T}[i,:]}^l = \widetilde{Q}_{\mathcal{T}[i,:]}^l / \|\widetilde{Q}_{\mathcal{T}[i,:]}^l\|_2 \tag{3}$$

$$\widetilde{Q}_{\mathcal{S}'}^{l'} = A_{\mathcal{S}'}^{l'} \cdot {A_{\mathcal{S}'}^{l'}}^{\top}; \quad Q_{\mathcal{S}'[i,:]}^{l'} = \widetilde{Q}_{\mathcal{S}'[i,:]}^{l'} / \|\widetilde{Q}_{\mathcal{S}'[i,:]}^{l'}\|_2 \tag{4}$$

where $\widetilde{Q}^l_{\mathcal{T}} \in \mathbb{R}^{B \times B}$ and $\widetilde{Q}^{l'}_{\mathcal{S'}} \in \mathbb{R}^{B \times B}$ denote *pairwise similarities* at teacher layer $l$ and student layer $l'$, $Q^l_{\mathcal{T'}}$ denotes the row-wise $L_2$ normalization of $\widetilde{Q}^l_{\mathcal{T}}$ and $Q^{l'}_{\mathcal{S'}}$ are the row-wise $L_2$ normalization of $\widetilde{Q}^l_{\mathcal{T}}$ and $\widetilde{Q}^{l'}_{\mathcal{S'}}$. In this way, the value scale of ANN and SNN similarity vectors can be normalized to the range [0, 1].

## 3.2 PAIRWISE-RELATIONAL LOSS

According to the definition of *pairwise similarities* above, there exists a potential problem when computing the similarities in SNNs. In the feature maps of SNNs, as the values of most features are often 0, the similarity vectors computed in SNNs may be very sparse, making the vector hard to match any similarity vectors of ANNs.

Considering this matching problem, we give a well-designed pairwise-relational loss function as

$$\mathcal{L}_{distill\_relation}(\mathcal{T}, \mathcal{S}) = \frac{1}{B} \sum_{i=1}^{B} \sum_{(l,l') \in \mathcal{I}} L(f(Q^l_{\mathcal{T}[i,:]}), f(Q^{l'}_{\mathcal{S'}[i,:]})) \tag{5}$$

where $f(\cdot)$ is a kernel function used to map similarity vectors into other feature spaces for separation, $\mathcal{I}$ is the set of layer pairs for loss calculation, $L(\cdot)$ is smoothL1 loss, which is defined as

$$L(Q^l_{\mathcal{T}[i,:]}, Q^{l'}_{\mathcal{S'}[i,:]}) = \begin{cases} \frac{1}{2}(f(Q^l_{\mathcal{T}[i,:]}) - f(Q^{l'}_{\mathcal{S'}[i,:]}))^2/\beta, & \text{if } |f(Q^l_{\mathcal{T}[i,:]}) - f(Q^{l'}_{\mathcal{S'}[i,:]})| < \beta \\ |f(Q^l_{\mathcal{T}[i,:]}) - f(Q^{l'}_{\mathcal{S'}[i,:]})| - \frac{1}{2} * \beta, & \text{otherwise.} \end{cases}$$
$$\tag{6}$$

where $\beta$ is the threshold at which to change between $L_1$ and $L_2$ loss. This smoothL1 loss is robust to the outlier vectors. If a sparse similarity vector of SNN makes the pairwise similarity distance larger than $\beta$, $L_1$ loss function is used to reduce the effect of the outlier input to the transfer learning; if the pairwise similarity distance is not larger than $\beta$, $L_2$ loss function is directly used to measure the learning loss of the input pair.

## 3.3 TRAINING

In this section, we provide a comprehensive description of our proposed *SuperSNN* for training SNNs.

**Training teacher ANNs.** We begin by training ANNs as pre-trained teacher models. From these teacher networks, we extract the outputs from their intermediate layers as *feature maps* and calculate the *pairwise similarities* that serve as guiding signals during the distillation process for training SNNs.

**Training student SNNs.** *SuperSNN* guides the training of a student SNN by incorporating an additional distillation loss. The loss function of *SuperSNN* to train the student SNN is formulated as follows:

$$L_{SuperSNN} = L_{CE} + \alpha * L_{distill} \tag{7}$$

where $L_{CE}$ denotes the entropy loss of the student SNN, $L_{distill}$ is the distillation loss computed from matching the feature vectors between the teacher ANN and the student SNN, and $\alpha$ is a hyperparameter.

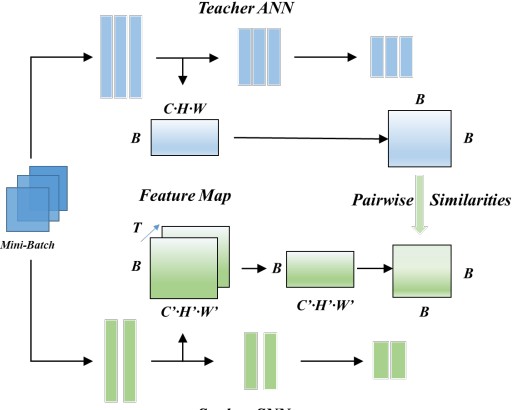

Figure 3: Illustration of generating the *pairwise similarities* of SNN. Using average pooling to get rid of the time dimension of SNN's feature maps and calculating the pairwise similarities of ANN and SNN.

**Backward propagation of SNNs.** In the error backpropagation, the classical backpropagation algorithm cannot be directly applied due to the non-differentiable nature of the spike activity function in equation 2. To address this issue, most previous works exploit surrogate gradients for the spatio-temporal backpropagation algorithms(Yujie

et al., 2018; Neftci et al., 2019). In this study, we employ a threshold-dependent batch normalization method (Zheng et al., 2020) to train SNNs, which uses the rectangular function(Yujie et al., 2018) to approximate the derivative of spiking activity, enabling the direct training of SNNs from a shallow structure (less than 10 layers) to a deep structure (50 layers). The pseudocode for the overall training process of *SuperSNN* is summarized in **Algorithm** 1.

---

**Algorithm 1** *SuperSNN*

---

**Require:** the SNN model $\mathcal{S}$, pre-train ANN model $\mathcal{T}$, input mini-batch $x$, true labels $y_{true}$, feature set $S_f = \emptyset$.
**Ensure:** the SNN model with knowledge from the ANN
 1: *# Forward propagation*
 2: **for** $(l, l')$ in $\mathcal{I}$ **do**
 3:     *# Get teacher feature maps*
 4:     $A_{\mathcal{T}}^l = \mathcal{T}(x)$    $A_{\mathcal{T}}^l \in \mathbb{R}^{B \times C \times H \times W}$
 5:     *# Get student feature maps*
 6:     $A_{\mathcal{S}}^{l'} = \mathcal{S}(x)$    $A_{\mathcal{S}}^{l'} \in \mathbb{R}^{B \times T \times C' \times H' \times W'}$
 7:     *# Average pooling the student feature maps*
 8:     $A_{\mathcal{S}'}^{l'} = \sum_{t=0}^{T} A_{\mathcal{S}}^{l'}/T$    $A_{\mathcal{S}'}^{l'} \in \mathbb{R}^{B \times C' \times H' \times W'}$
 9:     Add $(A_{\mathcal{T}}^l, A_{\mathcal{S}'}^{l'})$ to $S_f$
10: **end for**
11: *# Calculate the distillation loss*
12: Using equation 5, calculate $\mathcal{L}_{distill\_relation}$ with $S_f$
13: *# Calculate the total loss*
14: $L_{SuperSNN} = L_{CE} + \alpha * L_{distill}$
15: *# Backward propagation*
16: Calculate the gradients
17: Update parameters

---

# 4 EXPERIMENTS

## 4.1 EXPERIMENT SETTINGS

**Datasets.** We evaluated our proposed *SuperSNN* method on three datasets, including **CIFAR10** (Lecun & Bottou, 1998), **CIFAR100** (Krizhevsky & Hinton, 2009) and **Tiny ImageNet**. **CI-FAR10/CIFAR100** contain 60k RGB images (size $32 \times 32 \times 3$) in 10/100 categories, which are divided into 50k training samples and 10k testing samples. **Tiny ImageNet** contains 110k RGB images (size $64 \times 64 \times 3$) in 200 classes, which is a subset of ILSVRC2012. Each class includes 500 training samples and 50 testing samples.

**Backbone Architectures.** We employed six representative architectures as teacher ANNs to evaluate the performance of *SuperSNN*, including ResNet19 (He et al., 2016), ResNet34, Pyramidnet110 (Han et al., 2016), Pyramidnet50, WideResNet28 (Zagoruyko & Komodakis, 2016b), and Vgg16 (Simonyan & Zisserman, 2014); and employed ResNet19 (Sengupta et al., 2018) and Pyramidnet50 as student SNNs (More details in Appendix C.1).

**Implementation Details.** All experiments were conducted on one NVIDIA A100 GPU with 80GB memory. For teacher ANNs, the epoch number was set to 100, 200, 200 for **CIFAR10**, **CIFAR100** and **Tiny ImageNet** respectively, and the batch size we set as 64, 128 for **CIFAR** and **Tiny ImageNet**. We adopted the SGD optimization algorithm with an initial learning rate of 0.025, which decayed to 0.0025 when the training process reached its halfway point. For PyramidNet110 and PyramidNet50, the widening factor $\alpha_w$ and output feature dimension were both set to 270 and 286. During the training of SNNs, the epoch number was set to 100, 200, 200 for **CIFAR10**, **CIFAR100** and **Tiny ImageNet** respectively, and the batch size we set as 64. The hyperparameter $\alpha$ of feature-based and relation-based methods were set to 1,000 for **CIFAR10/CIFAR100**, and 200 for **Tiny ImageNet**. The time step was set to 2. The threshold $\beta$ in equation 6 was set to 1 and 3 for ResNet19 and PyramidNet respectively. We adopted the Adam optimization algorithm with an initial learning rate of 0.001, which decayed to 0.0001 when the training process reached its halfway

Table 1: Top-1 accuracy (%) of *SuperSNN* with existing methods on CIFAR10/CIFAR100. The best results (second best) are shown in **boldface** (underlined). Accuracy (%) of teacher ANNs: ResNet34/ResNet19: 96.15/95.30 on CIFAR10, 80.34/74.16 on CIFAR100; Pyramidnet110/Pyramidnet: 95.74/95.61 on CIFAR10, 80.59/78.58 on CIFAR100. Accuracy (%) of student SNNs: ResNet19: 92.15/70.51 on CIFAR10/CIFAR100, Pyramidnet50: 92.60/71.41 on CIFAR10/CIFAR100. * denotes the feature-based method(Zagoruyko & Komodakis, 2016a).

| Method | SNN | CIFAR10 Acc | CIFAR100 Acc | Time Step |
|---|---|---|---|---|
| Hybrid training (Rathi et al., 2020) | VGG11 | 92.22 | 67.87 | 125 |
| Diet-SNN (Rathi & Roy, 2020) | ResNet-20 | 92.54 | 64.07 | 10/5 |
| STBP (Yujie et al., 2018) | CIFARNet | 89.83 | - | 12 |
| TSSL-BP (Zhang & Li, 2020) | CIFARNet | 91.41 | - | 5 |
| STBP-tdBN (Zheng et al., 2020) | ResNet-19 | 92.92 | 70.86 | 4 |
| TET (Deng et al., 2022) | ResNet-19 | 94.44 | 74.72 | 6 |
| Rec-Dis (Guo et al., 2022) | ResNet-19 | 95.55 | 74.10 | 6 |
| Spikeformer (Zhou et al., 2023) | Spikformer-4-384 | 95.19 | 77.86 | 4 |
|  | Spikformer-4-384 400E | 95.51 | 78.21 | 4 |
| Response-based (Hinton et al., 2015) | ResNet34-ResNet19 | 92.85 | 75.76 | 2 |
|  | ResNet19-ResNet19 | 93.04 | 73.14 | 2 |
|  | Pyramidnet110-Pyramidnet50 | 93.00 | 76.60 | 2 |
|  | Pyramidnet50-Pyramidnet50 | 93.51 | 75.90 | 2 |
| Feature-based* | ResNet34-ResNet19 | 94.55 | 74.94 | 2 |
|  | ResNet19-ResNet19 | 94.40 | 75.55 | 2 |
|  | Pyramidnet110-Pyramidnet50 | 93.45 | 77.51 | 2 |
|  | Pyramidnet50-Pyramidnet50 | 93.60 | 76.44 | 2 |
| *SuperSNN* (ours) | ResNet34-ResNet19 | **95.61** | 77.45 | 6 |
|  | ResNet34-ResNet19 | 95.08 | 76.49 | 2 |
|  | ResNet19-ResNet19 | 95.03 | 75.60 | 2 |
|  | Pyramidnet110-Pyramidnet50 | 95.53 | **79.41** | 2 |
|  | Pyramidnet50-Pyramidnet50 | 95.59 | 78.41 | 2 |

point. For PyramidNet50, the widening factor $\alpha_w$ and output feature dimension were set to 270 and 286. Additionally, we empirically define the function in equation 5 as $f(x) = 2e^x$.

## 4.2 LEARNING FROM THE HOMOGENEOUS ANNS

We chose ResNet19 and Pyramidnet50 as student SNNs to evaluate the performance of *SuperSNN* with knowledge from the homogeneous ANNs. This includes scenarios where the student and teacher networks share the same depth or have the same block structure (but different depths). We tested cases in which teacher ANNs transfer knowledge in different forms (response-based knowledge, feature-based knowledge and relation-based knowledge) to student SNNs and compared them with current leading SNN approaches.

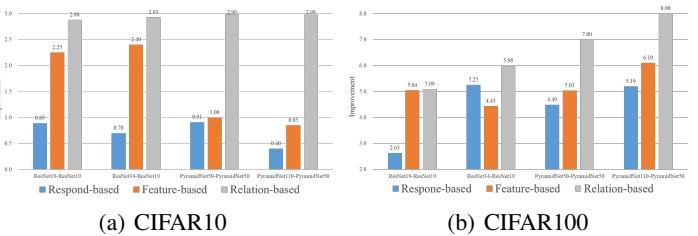

(a) CIFAR10        (b) CIFAR100

Figure 4: Improvement (%) of different knowledge distillation-based methods on CIFAR10 and CIFAR100. Each subfigure has described the results of different teacher-student groups including ResNet19-ResNet19, ResNet34-ResNet19, Pyramidnet50-Pyramidnet50 and Pyramidnet110-Pyramidnet50.

Experimental results on two benchmarks are summarized in Tables 1. It is observed that our proposed *SuperSNN* consistently improved the performance of student SNNs, enabling them to achieve significant accuracy compared with existing best methods. In Figure 4, the improvements of different knowledge distillation-based methods compared to student SNNs are depicted. Comparing the

results of response-based and feature-based approaches with those of *SuperSNN* in tables, it is clear that the latter consistently outperforms the former approaches on datasets **CIFAR10/CIFAR100**. This observation indicates the superiority of learning with knowledge derived from intermediate layers. Furthermore, we have conducted an analysis of the test accuracy curves for SNNs, and student SNNs when guided by homogeneous teacher ANNs using different distillation methods. From Figure 5 (a)-(d), we can see that *SuperSNN* plays a vital role in accelerating the convergence of student SNNs and helping them to achieve superior results. It's noteworthy that our proposed *SuperSNN* stands out as the best performer in terms of image classification. This observation suggests that *SuperSNN* is not only universal but also highly effective in enhancing the classification performance of student SNNs.

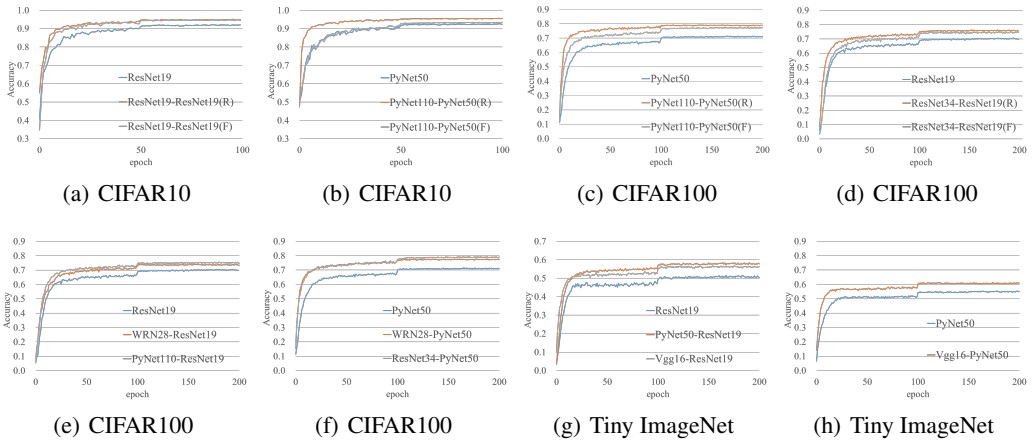

Figure 5: (a)-(d) Test accuracy curves of SNNs, and student SNNs during the training period under the guide of homogeneous teacher ANNs). (R) denotes the relation-based distillation method and (F) denotes the feature-based distillation method. (e)-(h) Test accuracy curves of SNNs, and student SNNs during the training period under the guide of heterogeneous teacher ANNs.

### 4.3    LEARNING FROM THE HETEROGENEOUS ANNS

To showcase the performance of knowledge transfer between the heterogenous teacher ANNs and student SNNs, we took ResNet34/19, PyramidNet110/50, WideResNet28 and Vgg16 as ANN teachers and tried to improve the image classification performance of PyramidNet50 and ResNet19 on datasets **CIFAR100** and **Tiny ImageNet**. We adopted *SuperSNN* for this study. Experimental results are shown in Table 2.

Table 2: Top-1 accuracy (%) and improvement (%) of *SuperSNN* with 2 time steps on CIFAR100 and Tiny ImageNet. The best results are shown in **boldface**.

|  | ANN-model | SNN-model | ANN | SNN | *SuperSNN* | Improvement |
|---|---|---|---|---|---|---|
| CIFAR100 | ResNet34 |  | 80.34 |  | 79.25 | 7.84 |
|  | ResNet19 | PyramidNet50 | 74.16 | 71.41 | 78.80 | 7.39 |
|  | WideResNet28 |  | 76.60 |  | 77.69 | 6.28 |
|  | Pyramidnet 110 |  | 80.59 |  | 75.59 | 5.08 |
|  | Pyramidnet 50 | ResNet19 | 78.58 | 70.51 | 74.40 | 3.89 |
|  | WideResNet28 |  | 76.60 |  | 74.14 | 3.63 |
| Tiny ImageNet | Vgg16 | PyramidNet50 | 56.10 | 55.37 | 61.43 | 6.06 |
|  | PyramidNet110 |  | 65.96 |  | 60.55 | 9.04 |
|  | PyramidNet50 | ResNet19 | 63.73 | 51.51 | 58.37 | 6.86 |
|  | Vgg16 |  | 56.10 |  | 59.49 | 7.98 |

From Table 2, it's evident that the classification performance of student SNNs has also improved significantly when learning knowledge from heterogeneous ANNs, enabling SNNs to achieve competitive or even superior results. More specifically, the top-1 accuracy of student PyramidNet50,

Table 3: Top-1 accuracy (%) of different loss functions for *SuperSNN* on three benchmarks.

| | ANN-model | SNN-model | $SuperSNN_{mse}$ | $SuperSNN_{smoothL1}$ |
|---|---|---|---|---|
| CIFAR10 | ResNet34 | ResNet19 | 94.84 | 95.08 |
| | ResNet19 | ResNet19 | 94.68 | 95.03 |
| | PyramidNet110 | PyramidNet50 | 95.38 | 95.58 |
| | PyramidNet50 | PyramidNet50 | 95.51 | 95.59 |
| CIFAR100 | ResNet19 | PyramidNet50 | 78.54 | 78.80 |
| | ResNet34 | ResNet19 | 75.88 | 76.49 |
| | WideResNet28 | PyramidNet50 | 77.56 | 77.69 |
| | Pyramidnet 110 | ResNet19 | 75.43 | 75.59 |
| Tiny ImageNet | Vgg16 | PyramidNet50 | 57.74 | 61.43 |
| | Vgg16 | ResNet19 | 56.72 | 59.49 |

with ResNet19 as its teacher, reaches 78.80% on **CIFAR100**, showing a notable 4.64% improvement compared to its teacher. We also have conducted an analysis of the test accuracy curves for SNNs, and student SNNs guided by heterogeneous teacher ANNs using the relation-based distillation method. As depicted in Figure 5 (e)-(h), we can see that learning with knowledge from heterogeneous teacher ANNs can also help the accuracy of student SNNs rise quickly during the training period.

Comparing the results on **CIFAR100** in Table 1 and Table 2, we observe that under the guidance of homogeneous teacher ANNs, student ResNet19 and Pyramidnet110 achieve the best results at 76.49% and 79.41%, respectively. These performances are superior to those achieved under the guidance of Pyramidnet, which are 75.59% and 79.25% for ResNet19 and Pyramidnet110, respectively. It seems that student SNNs could achieve better results with the help of their homogeneous teacher ANNs in terms of their top-1 accuracy on **CIFAR100**. Furthermore, it can be observed from the tables that a better student SNN still outperforms other students with the guidance of teacher ANNs.

## 4.4 DISCUSSIONS

**Loss function** To investigate the effectiveness of utilizing the defined pairwise-relational loss function as opposed to the MSE loss for our study, we denoted the proposed method employing the smoothL1 loss as $SuperSNN_{smoothL1}$ and conducted experiments to evaluate the impact of this choice by comparing it with the MSE loss approach, denoted as $SuperSNN_{mse}$. The detailed results, presented in Table 3, clearly demonstrate that $SuperSNN_{smoothL1}$ consistently outperforms $SuperSNN_{mse}$ on all benchmark datasets. This is particularly noticeable in the performance of student SNNs on **Tiny ImageNet**.

**Time Step** We conducted experiments to evaluate the top-1 accuracy of *SuperSNN* with different time steps on **CIFAR10/CIFAR100**. Experimental results are summarized in the Table 4. We can observe that the results demonstrate that the performance of *SuperSNN* exhibits notable improvement with only 2 time steps, and this performance is further enhanced with an increase in time steps. More specifically, student Pyramidnet50 learning from teacher ResNet34 with $T = 2$ achieve

Table 4: Top-1 accuracy (%) of *SuperSNN* with different time steps on CIFAR10/CIFAR100.

| ANN-model | SNN-model | CIFAR10 | CIFAR100 | Time Step |
|---|---|---|---|---|
| Pyramidnet110 | Pyramidnet50 | 95.58 | 79.46 | 6 |
| | | 95.55 | 79.37 | 4 |
| | | 95.53 | 79.14 | 2 |
| ResNet34 | Pyramidnet50 | 95.99 | 79.85 | 6 |
| | | 95.85 | 79.72 | 4 |
| | | 95.77 | 79.25 | 2 |
| ResNet19 | ResNet19 | 95.39 | 77.14 | 4 |
| | | 95.03 | 75.60 | 2 |
| Pyramidnet110 | ResNet19 | 94.68 | 76.41 | 4 |
| | | 94.42 | 75.59 | 2 |

95.77% and 79.25% on **CIFAR10/CIFAR100**, 0.22% and 0.60% lower than networks with T=4 respectively.

## 5 CONCLUSION

In this work, we proposed a novel T-S learning approach named *SuperSNN*, to guide SNNs learning with comprehensive supervisory information from ANNs. *SuperSNN* is a relation-based approach of knowledge distillation, in which the semantic relationship of the knowledge learned by ANN can be well preserved and transferred to SNN. In particular, we present an indirect matching approach, which unifies the value scale of SNN and ANN representation vectors and defines a pairwise-relational loss function, to alleviate the precision loss. To show the effectiveness of *SuperSNN*, we chose ResNet19 and Pyramidnet50 as SNN models and conducted comparisons *SuperSNN* with current leading SNN approaches over three benchmarks. Experimental results demonstrate no matter whether homogeneous or heterogeneous teacher ANNs are used, the proposed *SuperSNN* can outperform other SNN training methods using only two time steps for image classification, which proves the reliability and validity of *SuperSNN*.

There are several promising directions for future research that are worth exploring. Firstly, we see potential in applying *SuperSNN* to enhance the performance of larger models, such as Spikeformer (Zhou et al., 2023). This extension may promote the development of more complex and capable SNNs, pushing the boundaries of their applications. Furthermore, we also have an interest in exploring methods to improve the performance of SNNs on neuromorphic datasets, such as DVS128 Gesture (Amir et al., 2017).

### AUTHOR CONTRIBUTIONS

If you'd like to, you may include a section for author contributions as is done in many journals. This is optional and at the discretion of the authors.

### ACKNOWLEDGMENTS

Use unnumbered third level headings for the acknowledgments. All acknowledgments, including those to funding agencies, go at the end of the paper.

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

# A RELATED WORK

## A.1 LEARNING METHOD OF SPIKING NEURON NETWORKS

Current deep SNN training methods can be broadly categorized into two main classes: indirectly supervised learning algorithms, represented by ANN-SNN conversion (Bodo et al., 2017; Fang et al., 2021), and directly supervised algorithms, represented by spatio-temporal back-propagation (Yujie et al., 2018; Shrestha & Orchard, 2018).

ANN-to-SNN conversion can be seen as the most popular way to train SNNs in the past few years. It pre-trains a source ANN and then converts it to an SNN by changing the artificial neuron model to the spike neuron model (Baig, 2015). The basic idea is to use the firing rates (Han et al., 2020) or average postsynaptic potentials (Deng & Gu, 2021) of an SNN under the rate-coding scheme to approximate a ReLU-based ANN. Although ANN-to-SNN conversion is an effective method to obtain deep SNNs, it tends to overlook the rich temporal dynamic characteristics inherent to SNNs. Furthermore, it requires longer time steps to approach the accuracy of pre-trained ANNs (Rueckauer et al., 2016). This increases the SNN's latency, which can limit its practical applicability.

The directly supervised algorithm trains SNNs by unfolding the network over the temporal domains and computing the gradient on both spatial and temporal domains. Due to the non-differentiability of the spikes in SNNs, the surrogate gradient has been studied to implement spatio-temporal back-propagation by approximating the gradient with smooth functions (Yu et al., 2022). As the depth of the network increases, the vanishing gradient problem in SNN becomes more pronounced. To address this issue, several research studies have introduced novel normalization methods, the most well-known of which is tdBN(Zheng et al., 2020). It's worth noting that a directly supervised algorithm requires far fewer time steps compared to ANN-SNN conversion. Although extending the length of time steps contributes to more reliable gradients of surrogate functions (Yujie et al., 2018; Neftci et al., 2019; Zenke & Vogels, 2020), the primary drawback of a directly supervised algorithm is its inherent limitation in achieving optimal performance.

## A.2 KNOWLEDGE DISTILLATION

Knowledge distillation(Hinton et al., 2015) is a model compression method, which transfers the information from a large model or an ensemble of models into training a small model without a significant drop in accuracy.

There are three main knowledge types for knowledge distillation: response-based knowledge, feature-based knowledge and relation-based knowledge. Response-based knowledge usually refers to the neural response of the last output layer of the teacher model (Ba & Caruana, 2014; Hinton et al., 2015; Kim et al., 2018; Mirzadeh et al., 2020). The main idea is to directly mimic the final prediction of the teacher model. Feature-based knowledge from the intermediate layers is a good extension of response-based knowledge, *i.e.*, feature maps. The intermediate representations were first introduced in Fitnets(Romero et al., 2015), to provide hints to improve the training of the student model. The main idea is to directly match the feature activations of the teacher and the student. Inspired by this, a variety of other methods have been proposed to match the features indirectly (Romero et al., 2015; Zagoruyko & Komodakis, 2016a; Huang & Wang, 2017; Ahn et al., 2019; Heo et al., 2019). Relation-based knowledge further explores the relationships between different layers or data samples. (Yim et al., 2017) proposed a flow of solution process (FSP), which is defined by the Gram matrix between two layers. singular value decomposition was proposed to extract key information in the feature maps(Lee et al., 2018).

Recently, there has been a growing interest in applying knowledge distillation techniques to the field of SNN. Several studies have focused on leveraging the response-based knowledge obtained from a teacher ANN to enhance the classification performance of a student SNN(Xu et al., 2023a; Dong et al., 2023). Similarly, other works have focused on transferring the feature-based knowledge from the teacher ANN to the student SNN(Xu et al., 2023b; Kundu et al., 2021; Yang et al., 2022). However, the potential of exploring the untapped area of relation-based knowledge from the teacher ANN remains largely unexplored in the context of student SNN learning. This gap in research motivated our work to delve into this area and investigate its potential benefits.

## B    MORE DETAILS ABOUT THE FEATURE-BASED DISTILLATION METHOD

In this section, we describe the feature-based knowledge distillation method in detail. Figure 6 illustrates the overall procedure. Following (Zagoruyko & Komodakis, 2016a), we can derive *attention*

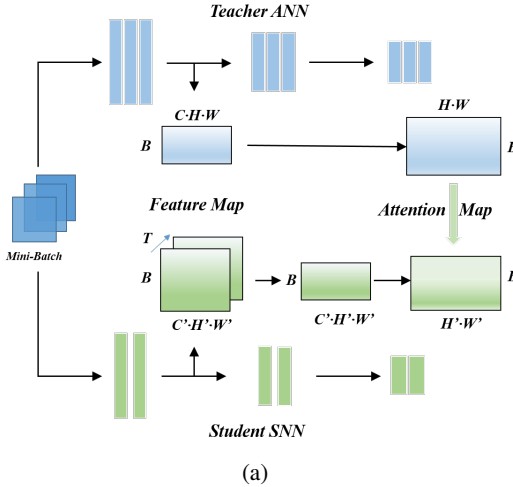

(a)

Figure 6:    Illustration of learning processes of the feature-based distillation method.

*maps* from *feature maps* to express knowledge of ANN and SNN, and then encourage the student SNN to produce similar normalized *attention maps* as the teacher ANN, which can be formulated as follows

$$F_{\mathcal{T}}^{l} = \frac{1}{C}\sum_{i=1}^{C}|A_{\mathcal{T}[:,i,:,:]}^{l}|^{2}; \quad F_{\mathcal{S}'}^{l'} = \frac{1}{C}\sum_{i=1}^{C}|A_{\mathcal{S}'[:,i,:,:]}^{l'}|^{2} \tag{8}$$

where $F_{\mathcal{T}}^{l} \in \mathbb{R}^{B \times H \times W}$ and $F_{\mathcal{S}'}^{l'} \in \mathbb{R}^{B \times H \times W}$ denote $(l, l')$ pair of teacher and student attention maps, respectively. The notation $[:, i, :, :]$ denotes the $i$th channel in the matrix. This distillation loss can be formulated by

$$\mathcal{L}_{distill\_feature}(F_{\mathcal{T}}, F_{\mathcal{S}'}) = \frac{1}{B}\sum_{i=1}^{B}\sum_{(l,l')\in\mathcal{I}}||\frac{F_{\mathcal{S}'[i,:,:]}^{l'}}{||F_{\mathcal{S}'[i,:,:]}^{l'}||_{2}} - \frac{F_{\mathcal{T}[i,:,:]}^{l}}{||F_{\mathcal{T}[i,:,:]}^{l}||_{2}}||_{2}. \tag{9}$$

where the notation $[i, :, :]$ denotes the $i$th data sample, $\mathcal{I}$ is the set of $(l, l')$ layer pairs, i.e., layers at the end of the same block.

Note that the height and width of the attention map $F_{\mathcal{T}}^{l}$ have to equal that of $F_{\mathcal{S}}^{l'}$. *SuperSNN* only requires that the ANN model and the SNN model share the same batch size. We consider it to be a more generalized approach for transferring knowledge between ANN and SNN.

## C    EXPERIMENT DETAILS

### C.1    MORE DETAILS ABOUT ARCHITECTURES OF DIFFERENT ANNS

In general, we chose six teacher ANNs in our experiments. For datasets **CIFAR10**, **CIFAR100**, and **Tiny ImageNet**, the classes of the final fully-connected layers are 10, 100, and 200 respectively.

For ResNets, the architecture details are shown in the following Table 5:

The only difference between our ResNet and ResNet(He et al., 2016) is the additional linear layer, which is the same as the ResNet-19Sengupta et al. (2018).

For Pyramidnets, the architecture details are shown in the following Table 6:

Table 5: Structure of ResNet

| layer name | Output size | | ResNet19 | | ResNet34 | |
| --- | --- | --- | --- | --- | --- | --- |
| | CIFAR | Tiny ImageNet | | | | |
| conv1 | 16*16 | 112*112 | 3*3, 64, stride 2 | | | |
| conv2_x | 8*8 | 56*56 | $\begin{bmatrix} 3*3, 64 \\ 3*3, 64 \end{bmatrix}$ | $*2$ | $\begin{bmatrix} 3*3, 64 \\ 3*3, 64 \end{bmatrix}$ | $*3$ |
| conv3_x | 4*4 | 28*28 | $\begin{bmatrix} 3*3, 128 \\ 3*3, 128 \end{bmatrix}$ | $*2$ | $\begin{bmatrix} 3*3, 128 \\ 3*3, 128 \end{bmatrix}$ | $*4$ |
| conv4_x | 2*2 | 14*14 | $\begin{bmatrix} 3*3, 256 \\ 3*3, 256 \end{bmatrix}$ | $*2$ | $\begin{bmatrix} 3*3, 256 \\ 3*3, 256 \end{bmatrix}$ | $*6$ |
| conv5_x | 1*1 | 7*7 | $\begin{bmatrix} 3*3, 512 \\ 3*3, 512 \end{bmatrix}$ | $*2$ | $\begin{bmatrix} 3*3, 512 \\ 3*3, 512 \end{bmatrix}$ | $*3$ |
| average pool | 1*1 | | AdaptiveAvgPool(1*1) | | | |

Table 6: Structure of PyramidNet

| layer name | Output size | | PyramidNet | |
| --- | --- | --- | --- | --- |
| | CIFAR | Tiny ImageNet | | |
| conv1 | 32*32 | 224*224 | $[3*3, 16]$ | |
| conv2_x | 32*32 | 224*224 | $\begin{bmatrix} 3*3, 16 + \alpha_w(k-1)/N \\ 3*3, 16 + \alpha_w(k-1)/N \end{bmatrix}$ | $*N_2$ |
| conv3_x | 16*16 | 112*112 | $\begin{bmatrix} 3*3, 16 + \alpha_w(k-1)/N \\ 3*3, 16 + \alpha_w(k-1)/N \end{bmatrix}$ | $*N_3$ |
| conv4_x | 8*8 | 56*56 | $\begin{bmatrix} 3*3, 16 + \alpha_w(k-1)/N \\ 3*3, 16 + \alpha_w(k-1)/N \end{bmatrix}$ | $*N_4$ |
| average pool | 1*1 | | AdaptiveAvgPool(1*1) | |

$\alpha_w$ denotes the widening factor and is set to 270. $N$ is the total number of layers(50/110 for PyramidNet50/PyramidNet110). $N_i$ is the number of layers for different layers, in our experiments, we set them to the same value $(N-2)/3$. $k$ is the current layer, its value range is $[1, N+1]$.

For WideResNet, the architecture details are shown in the following Table 7:

Table 7: Structure of wide residual networks

| layer name | Output size | WideResNet | |
| --- | --- | --- | --- |
| conv1 | 32*32 | $[3*3, 16]$ | |
| conv2_x | 32*32 | $\begin{bmatrix} 3*3, 16*k \\ 3*3, 16*k \end{bmatrix}$ | $*N_2$ |
| conv3_x | 16*16 | $\begin{bmatrix} 3*3, 32*k \\ 3*3, 32*k \end{bmatrix}$ | $*N_3$ |
| conv4_x | 8*8 | $\begin{bmatrix} 3*3, 64*k \\ 3*3, 64*k \end{bmatrix}$ | $*N_4$ |
| average pool | 1*1 | AdaptiveAvgPool(1*1) | |

The width of Wide residual networks is determined by factor $k$. $N_i$ is the number of layers for different layers, in our experiments, for wideresnet28, we set them to the same value 4.

Finally, we use the structure of Vgg16 (Simonyan & Zisserman, 2014) directly.

## C.2 MORE DETAILS ABOUT ARCHITECTURES OF DIFFERENT SNNS

We selected ResNet18 and PyramidNet50 as student SNNs. The architecture of the SNN models is identical to that of the ANN models, with the only difference being the utilization of the LIF neuron model and Threshold-dependent Batch Normalization (STBP-tdbn) (Zheng et al., 2020).

