# OpenReview forum: "SuperSNN: Training Spiking Neural Networks with Knowledge from Artificial Neural Networks"
_ICLR.cc/2024/Conference — ICLR 2024 Conference Withdrawn Submission_

### Official Review · Reviewer_e8ut · 2023-10-23

**Soundness:** 2 fair
**Presentation:** 2 fair
**Contribution:** 1 poor
**Rating:** 3
**Confidence:** 5

**Summary:**

This paper proposed a teacher-student learning in constructing deep SNN models which could avoid non-differentiable in spikes.Besides, this paper proposed an indirect relation-based approach, which defines a pairwise-relational loss function to build models.

**Strengths:**

Deep SNN construction is very important in AI areas. It is one of the basic rules to enhance performance in SNNs with direct or indirect training methods. Knowledge distillation in ANN-SNN is a promising way to build deep SNN models. This paper is based on the previous KD method to improve that named superSNN and evaluated the methods on CIFAR10, CIFAR100, Tiny ImageNet, experimental results show that the proposed methods could get general performance.

**Weaknesses:**

The biggest problem in this paper is the novelty is limited. This paper is based on some previous works about using kd between ANN-SNN [Xu et al., 2023a 2023b]. The argued relation based method is just another version of feature based and response based. Another dimension alignment was not solved in this paper, besides, the experimental results reported in this paper were not compared to the previous word in table 1, so I wonder about the detailed performance improvement which argued in this paper.

The organization of this paper is mixed, the authors should not put related work and some model details in the appendix section.

**Questions:**

Please see weakness.

---

### Official Review · Reviewer_L3Eg · 2023-10-28

**Soundness:** 2 fair
**Presentation:** 2 fair
**Contribution:** 1 poor
**Rating:** 3
**Confidence:** 5

**Summary:**

Inspired by the idea of Teacher–Student Learning, the authors provide a T-S method for SNN. Different from other T-S methods, this paper proposes an indirect relation-based approach. The paper did experiments on three image datasets.

**Strengths:**

Different from other T-S methods, this paper proposes an indirect relation-based approach.

**Weaknesses:**

1. The novelty is limited. This paper just transfers the widely used T-S method to SNN.
2. Many recent works are missing,
[1]  GLIF: A Unified Gated Leaky Integrate-and-Fire Neuron for Spiking Neural Networks
[2] Temporal Effective Batch Normalization in Spiking Neural Networks
[3] IM-Loss: Information Maximization Loss for Spiking Neural Networks

**Questions:**

1. Can you provide the experiments on Imagenet?
2. Can you provide ablation for the relation-based approach and logit-based approach with your code framework?
3. Can you provide the complete code in the Supplementary Material?

---

### Official Review · Reviewer_NQFs · 2023-10-28

**Soundness:** 2 fair
**Presentation:** 2 fair
**Contribution:** 2 fair
**Rating:** 3
**Confidence:** 5

**Summary:**

This paper proposes SuperSNN, which is a knowledge transfer learning framework. The core idea is to align the feature similarity matrix of the ANN and the SNN. In doing so, the problem of activation scale mismatch can be alleviated. The authors test this idea on three static image datasets and achieve some performance boost.

**Strengths:**

+ The idea of using a similarity matrix for knowledge distillation indeed solves the mentioned problem in the paper.

+ The framework is easy to implement.

**Weaknesses:**

major concerns:

- This paper proposes a knowledge distillation framework from ANN to SNN. The only contribution is the loss function used here is different from previous work. This contribution is not sufficient for conferences like ICLR.

- It is hard to determine that the usage of this loss function is unique in the literature. In the domain of knowledge distillation in ANNs, there are numerous works studying different types of loss functions. Whether this loss function is original remains questioned.

- I find the design choice of simply averaging the time dimension in SNN featuremaps inappropriate. The method in this paper is not the real way to compute the similarity matrix. Instead, to calculate the real similarity matrix of SNNs, the authors should flatten the SNN activation to $[B, TCHW]$ and then compute the $B\times B$ covariance matrix. For detailed explanation, check [1].

- Apart from accuracy, there are not many insightful discoveries for readers to understand the specific mechanism of this loss function.

- The experiments are not validated on ImageNet, which largely weakens the empirical contribution of this paper.

minor concerns:

- " The second is that the neuron state of ANNs is represented in binary format but that of SNNs is represented in float format, leading to the precision mismatch between ANNs and SNNs", wrong value formats of SNNs and ANNs.

- The related work should be placed in the main text, rather than the appendix. There are many spaces left on the 9th page.

[1]  Uncovering the Representation of Spiking Neural Networks Trained with Surrogate Gradient

**Questions:**

I am skeptical about the results of "response-based" KD. As I have tried these experiments on my own, the improvements are already impressive. Can the authors explain why the accuracy of response-based KD is only 92 on CIFAR-10? What is the baseline accuracy without any KD?

---

### Official Review · Reviewer_tGRN · 2023-10-30

**Soundness:** 3 good
**Presentation:** 1 poor
**Contribution:** 1 poor
**Rating:** 3
**Confidence:** 3

**Summary:**

The authors propose a new method to train spiking neural networks (SNNs). Their method is based on first training an artificial neural network (ANN) and then using intermediate feature representations of the ANN as a teacher signal for the SNN. The authors apply their method to several benchmark tasks and architectures and demonstrate the gains in performance over previous methods.

**Strengths:**

The authors apply their method to several datasets and use several different architectures, which demonstrates that their method is flexible and can indeed be applied to many different tasks. Their experimental results clearly show that their method indeed helps training SNNs and leads (overall) to improved performance.

**Weaknesses:**

The main issue in this paper is the premise of using ANNs as teachers, and the lack of motivation thereof. As the authors describe in their introduction, one of the main benefits of SNNs is their improved energy cost. However, if the method requires an ANN to be trained anyways, why would one even still have to train the SNN at all? Is the argument here purely about the cost of inference? While this might be valid in some (few) scenarios, in most cases the training cost by far outweighs the cost of inference. I believe that these points are problematic by themselves, but they are not even mentioned in the paper as limitations.

In addition, given that the paper allows using an entirely new source of information, the performance gains over previous methods are rather marginal. For example, on CIFAR10, their method seems to improve by about 1% over baseline methods, while requiring a full ANN to be trained.

Furthermore, the presentation of the paper could be improved. I believe that the paper could benefit from a more extensive section on background. Many important details are currently in the appendix, for example, what is relation-based knowledge? How does it differ from feature-based knowledge? These terms are even used in the overview figure 1, but are never described in the main paper.

Lastly, the authors propose to use average pooling to align the dimensions of the ANN and the SNN. This clearly leads to a loss of information of the SNN (many SNN activations can have the same average), but it is never explored nor discussed how this impacts performance.

**Questions:**

No questions.

---

### Official Review · Reviewer_L4ie · 2023-10-31

**Soundness:** 3 good
**Presentation:** 2 fair
**Contribution:** 2 fair
**Rating:** 5
**Confidence:** 3

**Summary:**

This paper introduces a method for training SNNs called SuperSNN, by using an ANN as a teacher within a Teacher-Student learning framework. In order to address the challenges of the additional temporal dimension and the binary representation of unit activity in SNNs, this work incorporates pairwise-relational knowledge distillation from ANNs into the loss function, alongside the entropy loss from the task. This study demonstrates that SuperSNN improves SNN performance, outperforming existing SNN training methods, and enabling the transfer of knowledge from both homogeneous and heterogeneous ANNs to SNNs.

**Strengths:**

- This paper presents an effective method for training SNNs, which leads to improved SNN performance.
- This paper demonstrates knowledge transfer that is not limited to ANNs and SNNs that share the same network architecture; it is also effective when the ANN and SNN have different network architectures.

**Weaknesses:**

- The figure legends, particularly for Figure 4 and Figure 5, are difficult to read and could be improved for clarity.
- I feel that this paper lacks a clear summary of existing efforts on knowledge distillation in SNNs, making the motivation and contributions of this work less clear. One suggestion I have is to move the related work section from the appendix to the main text. This would provide readers that have less background knowledge on this subject with a better understanding of how the current work connects and differs with the existing literature.
- This work shows a limited exploration of the temporal dimension in SNNs and focuses on processing static images, which overlooks the advantage of SNNs in processing temporal data.
- I would also appreciate a discussion of the limitations of the proposed approach.

**Questions:**

- The SNN models in this paper utilize a specific type of neuron model: the iterative LIF neurons. Can the proposed approach be extended to other neuron types, such as the Izhikevich neurons?
- It's not very clear to me what "time step" refers to in this study. What's the significance of achieving good performance with fewer time steps?